# Serum sCD40L and IL-31 in Association with Early Phase of IgA Nephropathy

**DOI:** 10.3390/jcm12052023

**Published:** 2023-03-03

**Authors:** Keiko Tanaka, Hitoshi Sugiyama, Hiroshi Morinaga, Masashi Kitagawa, Yuzuki Kano, Yasuhiro Onishi, Koki Mise, Katsuyuki Tanabe, Haruhito A. Uchida, Jun Wada

**Affiliations:** 1Department of Nephrology, Rheumatology, Endocrinology and Metabolism, Okayama University Faculty of Medicine, Dentistry and Pharmaceutical Sciences, Okayama 700-8558, Japan; 2Department of Medicine, Kawasaki Medical School General Medical Center and Department of Medical Care Work, Kawasaki College of Allied Health Professions, Okayama 700-8505, Japan; 3Department of Nephrology, National Hospital Organization Okayama Medical Center, Okayama 701-1192, Japan; 4Department of Chronic Kidney Disease and Cardiovascular Disease, Okayama University Faculty of Medicine, Dentistry and Pharmaceutical Sciences, Okayama 700-8558, Japan

**Keywords:** IgA nephropathy, cytokines, sCD40L, IL-31, proteinuria, inflammation, interstitial fibrosis

## Abstract

Background: IgA nephropathy (IgAN) is a major cause of chronic glomerulonephritis worldwide. T cell dysregulation has been reported to contribute to the pathogenesis of IgAN. Methods We measured a broad range of Th1, Th2 and Th17 cytokines in the serum of IgAN patients. We searched for significant cytokines, which were associated with clinical parameters and histological scores in IgAN patients. Results: Among 15 cytokines, the levels of soluble CD40L (sCD40L) and IL-31 were higher in IgAN patients and were significantly associated with a higher estimated glomerular filtration rate (eGFR), a lower urinary protein to creatinine ratio (UPCR), and milder tubulointerstitial lesions (i.e., the early phase of IgAN). Multivariate analysis revealed that serum sCD40L was an independent determinant of a lower UPCR after adjustment for age, eGFR, and mean blood pressure (MBP). CD40, a receptor of sCD40L, has been reported to be upregulated on mesangial cells in IgAN. The sCD40L/CD40 interaction may directly induce inflammation in mesangial areas and may therefore be involved in the development of IgAN. Conclusions: The present study demonstrated the significance of serum sCD40L and IL-31 in the early phase of IgAN. Serum sCD40L may be a marker of the beginning of inflammation in IgAN.

## 1. Introduction

IgA nephropathy (IgAN) is a major cause of chronic glomerulonephritis worldwide [1,2]. Approximately 40% of such patients develop end-stage renal disease (ESRD) within 20 years of the diagnosis [3]. The Oxford classification of pathologic features in IgAN has been proposed and internationally validated and is independently associated with the risk of disease progression [4,5,6]. An international risk-prediction model for disease progression that combines clinical data with pathologic features of IgAN in multiple ethnic groups has recently been reported [7].

Another characteristic of IgAN is that it is an autoimmune disease that is based on the binding of the glycan-specific IgG autoantibody to galactose-deficient IgA1 (Gd-IgA1) as an autoantigen [8,9]. Several pathogenic models of IgAN have been proposed. The dysregulation of the mucosal immune system in response to mucosal antigens results in mucosal B cell proliferation, leading to excessive B cell activating factor (BAFF) and a proliferation-inducing ligand (APRIL) signaling [10,11]. The T cell-dependent production of IgA is mainly stimulated by interleukin (IL)-6, IL-10, transforming growth factor (TGF)-β, BAFF and APRIL produced by intestinal epithelial, dendritic, and stromal cells [2]. Changes in circulating T cell subpopulations, including imbalance of helper T (Th) cells (e.g., Th1 and Th2) and the involvement of T cell cytokines in the posttranslational modification of the IgA1 hinge region may stimulate the production of Gd-IgA1 [12].

Renin-angiotensin blockade can, to some extent, reduce the level of proteinuria and the risk of renal failure in patients with high-risk IgAN. Despite this therapy, a large proportion of patients still develop ESRD. Corticosteroids and other immunosuppressants may be effective treatments for high-risk IgAN patients; however, recent trials did not draw any definitive conclusions [13,14].

To gain insight into the role of T cell cytokines in the disease activity and their potential role in the disease severity of IgAN, we measured a broad range of Th1, Th2 and Th17 cytokines in serum, which could lead to the exploration of novel biomarkers of the disease, as well as disease-specific therapies.

## 2. Materials and Methods

### 2.1. Study Population and Data Collection

The study group consisted of 114 patients with IgAN, 10 patients with autosomal dominant polycystic kidney disease (ADPKD), and 5 healthy subjects. This study focused on primary IgAN and excluded patients considered to have secondary IgAN complicated by gastrointestinal or liver disease, infection, malignancy, IgA vasculitis or other autoimmune abnormalities. Clinical data were obtained from the patients’ medical records. eGFR was calculated by using Modified for Japanese subjects: eGFR (mL/min/1.73 m^2^)  =  194 × serum Creatinine (mg/dL) ^−1.094^ × Age^−0.287^ (×0.739 for females) [15].

### 2.2. Ethical Issue

The protocol of the present study was approved by the Ethical Committee of the Okayama University Graduate School of Medicine, Dentistry and Pharmaceutical Sciences (Approval no. 1009). All subjects gave their written informed consent prior to participation in the study. The study was conducted in accordance with the Declaration of Helsinki.

### 2.3. Multiplex Assay for Cytokines

Cytokines in serum samples were measured by Bio-Plex Pro Human Th17 Cytokine Assays (Bio-Rad Laboratories, Inc., Tokyo, Japan), according to the manufacturer’s instructions. In brief, the samples were diluted 4-fold with the diluting solution and centrifuged at 10,000× *g* for 5 min. Fifty microliters of the supernatant were used for the following cytokine assays: IL-1β, IL-4, IL-6, IL-10, IL-17A, IL-17F, IL-21, IL-22, IL-23, IL-25, IL-31, IL-33, IFN-γ, soluble CD40 ligand (sCD40L), and TNF-α. The lower limits of detection, according to our standard curves, were 0.27–0.28 pg/mL for IL-1β, 1.3–4.3 pg/mL for IL-4, 0.9–2.6 pg/mL for IL-6, 2.0–3.7 pg/mL for IL-10, 1.5–2.0 pg/mL for IL-17A, 1.8–8.0 pg/mL for IL-17F, 5.0–20.5 pg/mL for IL-21, 4.6–5.0 pg/mL for IL-22, 7.0–26.1 pg/mL for IL-23, 1.2–1.4 pg/mL for IL-25, 2.8–4.8 pg/mL for IL-31, 3.3–7.8 pg/mL for IL-33, 3.2–3.9 pg/mL for IFN-γ, 3.2–7.4 pg/mL for sCD40L, and 0.3–2.4 pg/mL for TNFα.

### 2.4. Serum Gd-IgA1 Measurement

The serum levels of galactose deficient IgA1, a circulating pathogenic molecule in IgAN, were measured with an ELISA kit, according to the manufacturer’s instructions (Immuno-Biological Laboratories, Gunma, Japan), using KM55 monoclonal antibodies [16].

### 2.5. Histological Assessment

Histological findings were evaluated by two independent nephrologists as the MEST-C score according to the Oxford classification of IgAN [5,17,18]. The lesion scores were determined according to mesangial hypercellularity, assessed by the mesangial score (≤0.5 [M0] or >0.5 [M1]), endocapillary hypercellularity (absent [E0] or present [E1]), segmental glomerulosclerosis (absent [S0] or present [S1]), tubular atrophy and interstitial fibrosis (≤25% [T0], 26–50% [T1] or >50% [T2]), and cellular and fibrocellular crescents (absent [C0], present in at least 1 glomerulus [C1] or present in >25% of glomeruli [C2]). T1 and T2, C1, and C2 were combined, as reported previously [19]. Patients with IgAN who had <8 glomeruli in renal biopsy specimens were excluded from this study.

### 2.6. Statistical Analyses

The results are expressed as the mean ± SD or the median and interquartile range (IQR) for continuous data. *p* values of <0.05 were considered to indicate statistical significance. For multiple comparisons of the three groups of IgAN and ADPKD patients and healthy controls, every two groups were analyzed using Fisher’s exact test for categorical data and using Student’s *t*-test or the Mann–Whitney U test for continuous data, as appropriate. Bonferroni correction was then applied (*p* value of <0.016 was considered significant). Correlations among the cytokines and other variables in IgAN patients were evaluated by Spearman correlation analysis. Multivariate-adjusted regression analysis and multiple logistic regression analysis were performed to determine the factors that were significant independent predictors of proteinuria and renal function in IgAN patients. The statistical analyses were performed using the JMP software program (version 11, SAS Institute Inc., Cary, NC, USA).

## 3. Results

This study included 114 IgAN patients (male, 50%), with a mean age of 41.7 ± 15.5 years (range 16–78 years). All IgAN patients received renal biopsies. Their kidney function was relatively preserved, and they had various degrees of proteinuria and histological changes (Table 1). The five healthy controls were similar to the IgAN patients with regard to age, gender and eGFR. The 10 ADPKD patients were older and had lower eGFR values than the IgAN patients (Table 1).

Among the 15 cytokines examined in this study, the serum-soluble CD40 ligand (sCD40L) and IL-31 levels were extremely high and were detectable in almost all IgAN patients (Appendix A). In IgAN patients, both serum sCD40L and IL-31 were significantly higher in comparison to those in ADPKD patients or healthy controls (*p* < 0.0001 *) (Figure 1). However, since differences in age and renal function may affect serum cytokine levels, multivariate-adjusted regression analyses were performed to determine whether or not cytokines correlated with IgAN, using the age and eGFR for adjustment, in the cohort of IgAN and ADPKD patients and healthy controls. The results showed that IgAN (vs. not IgAN) was a significant factor explaining serum sCD40L (*p* < 0.0001 *) and IL-31 (*p* < 0.0001 *) levels (Appendix A). Therefore, in IgAN patients, both serum sCD40L and IL-31 were significantly higher than in ADPKD patients or healthy control.

Next, we focused on the population of IgAN patients. Univariate analysis of the IgAN patients revealed a significant negative correlation between serum sCD40L and the urinary protein to creatinine ratio (UPCR) (*p* = 0.0087, R^2^ = 0.059; Figure 2A). A significant positive correlation was found between serum sCD40L and the estimated glomerular filtration rate (eGFR) (*p* = 0.016, R^2^ = 0.050; Figure 2B). The same significant correlations were also observed between serum IL-31 and the UPCR and eGFR (Figure 2C,D). We found that the patients with higher UPCR values had lower eGFR values (*p* = 0.0007 *) in the IgAN patients. The associations between these cytokines and lower UPCR values may be confounded by the lower eGFR values in patients with high UPCR values, which were correlated with lower cytokines. Interestingly, a high positive correlation was observed between sCD40L and IL-31 (*p* < 0.0001, R^2^ = 0.68) (Figure 2E) in IgAN patients.

In a multivariate-adjusted regression analysis of the IgAN patients, after adjustment for age (Model 1), age and eGFR (Model 2), and age, eGFR and mean blood pressure (MBP) (Model 3), serum sCD40L was independently associated with a lower UPCR (Table 2). Serum sCD40L was significantly associated with lower odds of having proteinuria (≥1.0 g/gCr) in Model 3 (adjusted for age, eGFR and MBP) (Table 3). The multivariate odds ratio for sCD40L (per 100 pg/mL increase) was 0.79 (95%CI: 0.63–0.96, *p* = 0.032). On the other hand, after adjustment for age, eGFR, and MBP (Model 3), serum IL-31 was an independent determinant of the UPCR (≥1.0 g/gCr) (Table 4) but was not significantly associated with the continuous variable of the UPCR in the multivariate-adjusted regression analysis (Table 5). Serum sCD40L and IL-31 were not independently associated with eGFR in any of the multivariate-adjusted regression analysis models (Appendix A).

Based on the Oxford classification, both serum sCD40L and IL-31 were significantly higher in IgAN patients with milder tubular atrophy/interstitial fibrosis (lower T-score). Lesion scores other than the T-score were not associated with serum sCD40L or IL-31 (Table 6).

As previously reported [20], the serum Gd-IgA1 levels were positively correlated with serum IgA and the UPCR (Appendix A), but—contrary to our expectations—were not positively correlated with serum sCD40L and IL-31 (Appendix A) in IgAN patients.

TNF-*α* is a cytokine that was mildly increased and detectable in the serum of 98% of IgAN patients (Appendix A). Serum TNF-*α* had no significant association with the UPCR or eGFR (Appendix A) in IgAN patients. TNF-*α* was significantly associated with sCD40L and IL-31 (Appendix A), but not with Gd-IgA1 (Appendix A) in IgAN patients.

## 4. Discussion

The present study indicated that serum sCD40L and IL-31 levels were higher in IgAN patients and were significantly associated with a higher eGFR, lower UPCR, and milder tubulointerstitial lesions (i.e., the early phase of IgAN). Serum sCD40L was independently associated with lower UPCR values in IgAN patients.

It has been recently reviewed that IgAN is characterized by higher proportions of circulatory Th2, T follicular helper (Tfh), Th17, and Th22 cells but lower Th1 and Tregs [12]. The review report mentions that Th2, Th17 and Tfh cytokines contribute to the elevated synthesis of Gd-IgA1 and that the production of anti-Gd-IgA1 autoantibodies may be stimulated by Tfh cells. In the present study, IL-4 (a representative Th2 cytokine), IL-21 (a Tfh cytokine), IL-17 and IL-22 were not detectable in most IgAN patients. IL-6 (a Th2 cytokine) was detected and slightly increased in half of the IgAN patients. TNF-α (a Th1 cytokine) was detectable and mildly increased but was not associated with Gd-IgA1 or any clinical parameters.

CD40L is expressed after activation on all Th subsets (Th1, Th2, Tfh, Th17 cells) other than Tregs [12]. CD40L engages the B-cell receptor CD40 and induces B-cell growth, differentiation, and IgA class switching. IL-31 is another Th2 cytokine, which may be involved in the Gd-IgA1 synthesis as described above. The significant increase of sCD40L and IL-31 in serum may induce B cells to produce excessive Gd-IgA1, leading to IgA deposition in the glomeruli and glomerular injury [21]. However, this study indicated that sCD40L and IL-31 were not positively correlated with serum Gd-IgA1 levels or with the histological scores of glomerular lesions. This suggests that sCD40L and IL-31 have no direct contribution to the production of Gd-IgA1.

In particular, sCD40L was independently associated with lower UPCR values in IgAN patients. CD40L is expressed on activated platelets as well as activated Th cells [22]. Soluble CD40L (sCD40L) is rapidly released from those after activation and has been widely studied as a marker of inflammatory states and autoimmune diseases, including atherosclerosis, rheumatoid arthritis, and systemic lupus erythematosus [23,24,25]. No previous studies have investigated the significance of serum sCD40L in IgAN patients. This study indicates that sCD40L may be a marker of the beginning of inflammation in IgAN.

CD40, a receptor of CD40L, is expressed on antigen-presenting cells (APCs) such as B cells, monocytes, and dendritic cells. CD40/CD40L interaction has a critical role in many aspects of the immune response [22]. CD40/CD40L has been shown to cause the initiation and progression of renal diseases, such as membranous nephritis and lupus nephritis [26,27], which occur due to B cell activation. In the kidney, CD40 expression is induced in mesangial cells, podocytes, and tubular epithelial cells by various pathological stimuli [28,29,30]. The upregulation of CD40 on mesangial cells has been observed in kidney biopsy specimens of IgAN patients [31]. The stimulation with sCD40L and other inflammatory cytokines leads to increased CD40 expression in podocytes and tubular epithelial cells [32,33]. Their upregulation of CD40 contributes to their inflammatory response and the following fibrotic process in kidney disease [28]. Increased sCD40 in serum may directly induce inflammation of mesangial cells in the early phase and may contribute to the progression of proteinuria and interstitial fibrosis in the following phase of IgAN. The reason for the ‘negative’ correlation between serum sCD40L and proteinuria is unclear. However, it may simply be due to the differences in the timing of T cell and platelet activation versus podocyte injury.

Another significantly increased cytokine, IL-31, is not reported to be associated with any autoimmune disease, including IgAN. IL-31 has recently been described as the main cytokine involved in allergies, such as cutaneous allergic reactions and asthma [34,35]. Allergies could be a cause of the development of IgAN but cannot explain all of the increased IL-31 in the serum of IgAN patients. An increase in IL-31 may accompany the significant increase of sCD40L because a high correlation was observed between serum sCD40L and IL-31 in IgAN patients. These cytokines are both Th2 cytokines, but their molecular connection is unknown. The same result has been reported in patients with relapsing-remitting multiple sclerosis (MS) [36]. IL-31 and sCD40L have been shown to be positively correlated with MS severity [37]. In the pathogenesis of MS, IL-31 and sCD40L have been reported to be secreted by activated mast cells, amplifying the T cell immune response in the central nervous system [37]. Mast cells have been suggested to be associated with the progression of interstitial fibrosis in IgAN [38,39], but their role in the early phase of IgAN is not clear. There may be co-stimulators or common pathways that increase the serum levels of both sCD40L and IL-31 but not mast cells in the pathogenesis of IgAN.

The detection of increased serum sCD40L in the early phase of IgAN may give new insight into the effectiveness of antiplatelet therapy. Platelets are also thought to be responsible for the initiation and progression of glomerular injury [40]. Platelets or platelet-releasing growth factors, such as platelet-derived growth factor (PDGF) and transforming growth factor-β (TGF-β), have been shown to induce mesangial proliferation and matrix accumulations [41]. Serum sCD40L may be mostly derived from activated platelets, and antiplatelet therapy (e.g., clopidogrel) is reported to reduce plasma sCD40L [42]. In a past meta-analysis, antiplatelet therapy was suggested to reduce proteinuria and protect against renal dysfunction in patients with IgAN [43]. A therapeutic approach that inactivates platelets may control the inflammation induced by sCD40L/CD40, especially in the early phase of IgAN.

The present study was associated with several limitations. First, we did not histologically analyze the levels of cytokines or their receptors, including IL-31 and sCD40L, in the kidney tissues of IgAN patients. Our preliminary study using assays that were used in this study indicated that these cytokines were not detectable or were at very low levels in urine samples; therefore, we did not evaluate the local levels of these cytokines. Further studies are necessary to determine the CD40 expression in mesangial cells and the surrounding cells, as well as the correlation with the CD40 expression and clinical parameters. Second, we did not compare the serum levels of these cytokines in IgAN to those in other autoimmune glomerulonephritides, such as lupus nephritis, and thus it is possible that the increased serum cytokines may not be specific to IgAN. Further studies are therefore necessary to clarify the local expression of these cytokines and their effects on disease progression in IgAN. Whether these cytokines are prognostic factors for early IgAN or not is particularly important and should be investigated in the prospective study in the next settings

In conclusion, serum sCD40L and IL-31 were increased and significantly associated with the early phase of IgAN. Serum sCD40L was independently associated with lower proteinuria and may be a marker of the beginning of inflammation in IgAN.

## Figures and Tables

**Figure 1 jcm-12-02023-f001:**
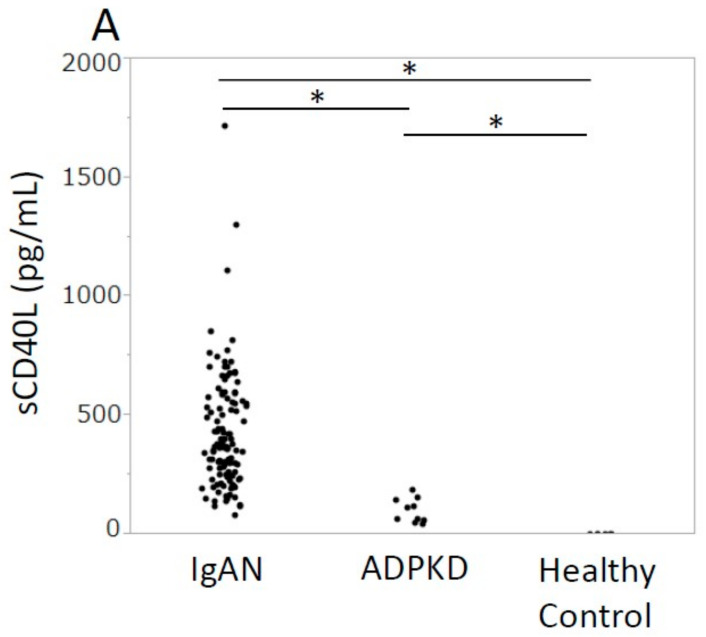
The distribution of serum sCD40L/IL-31 in three groups. The sCD40L (**A**) and IL-31 (**B**) levels in IgAN patients were significantly higher in comparison to ADPKD patients and healthy controls. ADPKD, autosomal dominant polycystic kidney disease; IgAN, IgA nephropathy; IL-31, interleukin 31; sCD40L, soluble CD40 ligand. * *p* < 0.016.

**Figure 2 jcm-12-02023-f002:**
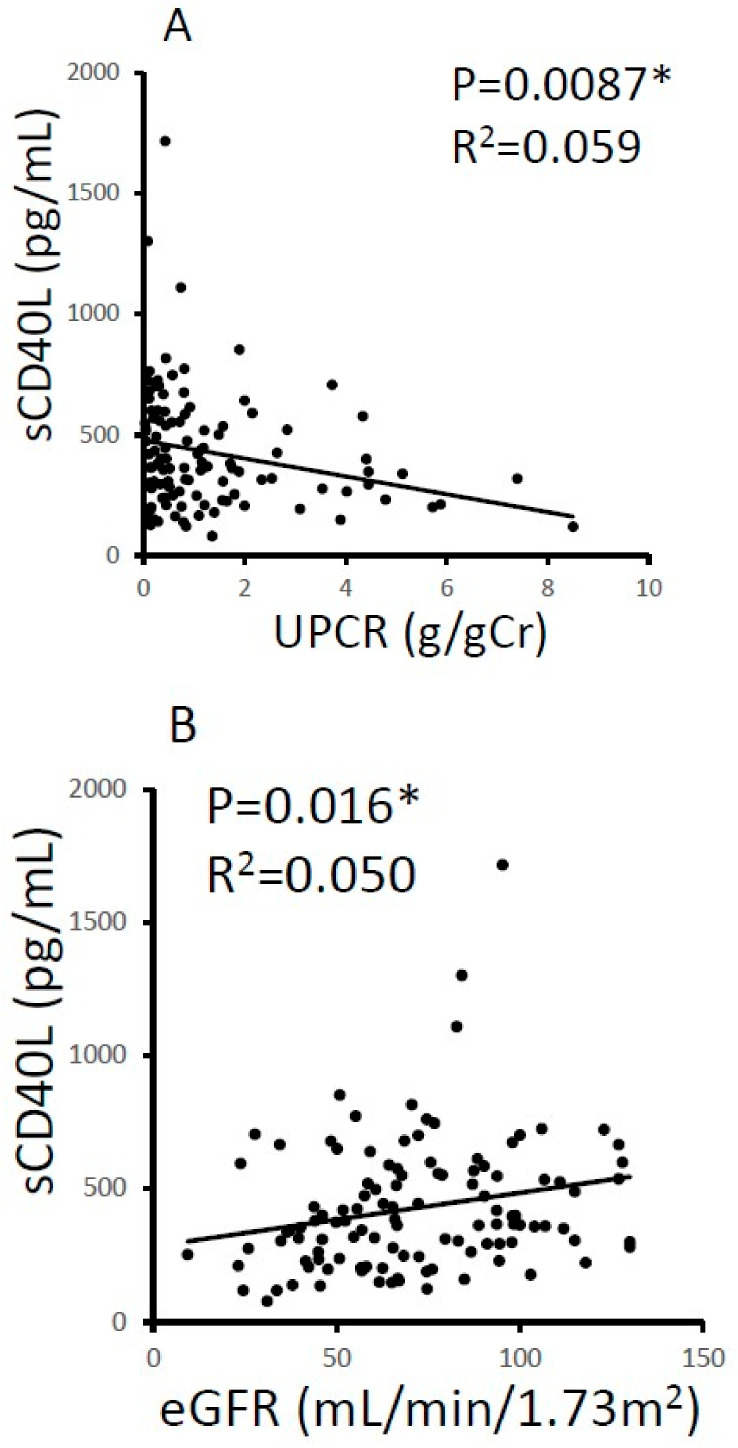
Relationship among serum sCD40L/IL-31, proteinuria (UPCR), and the renal function (eGFR) in IgA nephropathy patients. Serum sCD40L was significantly associated with a lower UPCR (**A**) and higher eGFR (**B**). Similar relations were found between IL-31 and the UPCR (**C**), and eGFR (**D**). Serum sCD40L was highly correlated with IL-31 (**E**). eGFR, estimated glomerular filtration rate; IL-31, interleukin 31; sCD40L, soluble CD40 ligand; UPCR, urinary protein to creatinine ratio. * *p* < 0.05.

**Table 1 jcm-12-02023-t001:** Baseline characteristics of the study subjects.

	IgAN	ADPKD	Healthy Control
*N*	114	10	5
Age (years)	41.7 ± 15.5 ^b^*	58.2 ± 16.8 ^a^*	37 ± 8.4
Gender, male, *n* (%)	57 (50)	5 (50)	3 (60)
eGFR (ml/min/1.73 m^2^)	71.4 ± 27.3 ^b^*	41.8 ± 29.2 ^ac^*	79.5 ± 8.08 ^b^*
UPCR (g/gCr)	0.73 (0.28–1.59) ^bc^*	0.09 (0.07–0.51) ^ac^*	0.03 (0.02–0.05) ^ab^*
Serum sCD40L (pg/mL)	367 (252–560) ^bc^*	86.9 (55.4–146) ^ac^*	0.0 ^ab^*
Serum IL-31 (pg/mL)	348 (182–543) ^bc^*	0.0 ^a^*	0.0 ^a^*
MBP (mmHg)	94.8 (87–103)	92.5 (74–101)	n.d.
Diabetes mellitus, *n* (%)	10 (8.8)	1 (10.0)	0 (0.0)
Antihypertensive drugs, *n* (%)	44 (30.1)	7 (70.0)	0 (0.0)
Oxford classification, *n* (%)			
M1	4 (3.5)	-	-
E1	33 (28.9)	-	-
S1	50 (43.9)	-	
T1-2	34 (29.8)	-	-
C1-2	75 (65.8)	-	-

Data are expressed as the mean ± SD, median (interquartile) or number (percentage). ADPKD, autosomal dominant polycystic kidney disease; eGFR, estimated glomerular filtration rate; IgAN, IgA nephropathy; MBP, mean blood pressure; SD, standard deviation; UPCR, urinary protein to creatinine ratio; n.d., not determined; M, mesangial hypercellularity; E, endocapillary hypercellularity; S, segmental glomerulosclerosis; T, tubular atrophy and interstitial fibrosis; C, cellular and fibrocellular crescents. Note that the details of the Oxford classification of IgAN described in Methods. ^a^*: *p* < 0.016 vs. IgAN, ^b^*: *p* < 0.016 vs. ADPKD, ^c^*: *p* < 0.016 vs. Healthy control.

**Table 2 jcm-12-02023-t002:** Multivariate-adjusted regression analysis of predictors of proteinuria in different models including serum sCD40L in IgAN.

	β	*p*	Model R^2^
Model 1 sCD40L ≥ median (pg/mL)	−0.37	0.011 *	0.11
Model 2 sCD40L ≥ median (pg/mL)	−0.35	0.016 *	0.13
eGFR < 60 (mL/min/1.73 m^2^)	0.28	0.086	
Model 3 sCD40L ≥ median (pg/mL)	−0.32	0.027 *	0.14
eGFR < 60 (mL/min/1.73 m^2^)	0.28	0.091	
MBP ≥ median (mmHg)	0.20	0.16	

Adjusted for age. The median values of serum sCD40L is 367 (pg/mL). eGFR, estimated glomerular filtration rate; IgAN, IgA nephropathy; MBP, mean blood pressure; sCD40L, soluble CD40 ligand. * *p* < 0.05.

**Table 3 jcm-12-02023-t003:** Multivariate odds ratios for proteinuria (≥1.0 g/gCr) in different models including serum sCD40L in IgAN.

	Odds Ratio	95% CI	*p*
Model 1 sCD40L (per 100 pg/mL)	0.77	0.61–0.93	0.016 *
Model 2 sCD40L (per 100 pg/mL)	0.78	0.63–0.95	0.023 *
eGFR (per 10 mL/min/1.73 m^2^)	0.87	0.71–1.05	0.17
Model 3 sCD40L (per 100 pg/mL)	0.79	0.63–0.96	0.032 *
eGFR (per 10 mL/min/1.73 m^2^)	0.88	0.71–1.08	0.23
MBP (per 10 mmHg)	1.07	0.77–1.52	0.65

Adjusted for age. CI, confidence interval; eGFR, estimated glomerular filtration rate; IgAN, IgA nephropathy; MBP, mean blood pressure; sCD40L, soluble CD40 ligand. * *p* < 0.05.

**Table 4 jcm-12-02023-t004:** Multivariate odds ratios for proteinuria (≥1.0 g/gCr) in different models including serum IL-31 in IgAN.

	Odds Ratio	95% CI	*p*
Model 1 IL-31 (per 100 pg/mL)	0.81	0.66–0.96	0.024 *
Model 2 IL-31 (per 100 pg/mL)	0.82	0.67–0.97	0.038 *
eGFR (per 10 mL/min/1.73 m^2^)	0.87	0.71–1.06	0.19
Model 3 IL-31 (per 100 pg/mL)	0.82	0.67–0.98	0.049 *
eGFR (per 10 mL/min/1.73 m^2^)	0.89	0.72–1.09	0.28
MBP (per 10 mmHg)	1.12	0.80–1.57	0.50

Adjusted for age. CI, confidence interval; eGFR, estimated glomerular filtration rate; IgAN, IgA nephropathy; IL-31, interleukin 31; MBP, mean blood pressure. * *p* < 0.05.

**Table 5 jcm-12-02023-t005:** Multivariate-adjusted regression analysis of predictors of proteinuria in different models including serum IL-31 in IgAN.

	β	*p*	Model R^2^
Model 1 IL-31 ≥ median (pg/mL)	−0.33	0.028 *	0.098
Model 2 IL-31 ≥ median (pg/mL)	−0.31	0.037 *	0.12
eGFR < 60 (mL/min/1.73 m^2^)	0.29	0.081	
Model 3 IL-31 ≥ median (pg/mL)	−0.29	0.051	0.14
eGFR < 60 (mL/min/1.73 m^2^)	0.28	0.089	
MBP ≥ median (mmHg)	0.24	0.10	

Adjusted for age. The median values of serum IL-31 is 348 (pg/mL). eGFR, estimated glomerular filtration rate; IgAN, IgA nephropathy; IL-31, interleukin 31; MBP, mean blood pressure. * *p* < 0.05.

**Table 6 jcm-12-02023-t006:** Serum levels of cytokines according to the Oxford classification of IgAN.

	sCD40L (pg/mL)				IL-31 (pg/mL)	
	Score 0	Score 1	*p*	Score 0	Score 1	*p*
M	367.9 (271.6–572.0)	267.7 (136.8–398.5)	0.332	352.9 (196.0–556.5)	214.6 (124.6–304.6)	0.232
E	367.6 (267.8–598.3)	380.5 (249.1–520.6)	0.511	361.1 (209.3–564.4)	304.1 (191.7–494.8)	0.310
S	425.4 (207.6–573.4)	363.6 (290.5–567.1)	0.696	361.1 (159.8–572.1)	321.9 (247.0–507.1)	0.671
	Score 0	Score 1–2	*p*	Score 0	Score 1–2	*p*
T	399.4 (295.2–595.9)	314.2 (206.7–473.2)	0.008 *	370.6 (248.1–562.7)	298.9 (136.8–396.5)	0.006 *
C	408.6 (318.4–660.0)	351.9 (230.6–551.4)	0.089	370.6 (279.3–569.2)	321.9 (182.3–522.5)	0.114

Data are expressed as the median (interquartile). M, mesangial hypercellularity; E, endocapillary hypercellularity; S, segmental glomerulosclerosis; T, tubular atrophy and interstitial fibrosis; C, cellular and fibrocellular crescents; IgAN, IgA nephropathy; IL-31, interleukin 31; sCD40L, soluble CD40 ligand. * *p* < 0.05.

## Data Availability

The cohort data used in this article contain anonymized but individual data. Therefore, we would prefer not to share this database.

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
