# Peer review of "Serum sCD40L and IL-31 in Association with Early Phase of IgA Nephropathy"

_jcm, 2023, doi:10.3390/jcm12052023_

Round 1

Reviewer 1 Report

1.     It is difficult to view the increase or decrease of cytokine as a result unique to igAN. This comparison cannot be explained by 10 ADPKD patients compared to the control group and 5 normal patients.

2.     Although the IgAN patient group is 114, there is no classification of comorbidity for secondary IgAN exclusion.

3.     Although the group of 114 patients who underwent renal biopsy was well collected, it would be meaningful to compare with a larger group of patients and an appropriate control group (normal adults, other chronic renal disease) to explain the change by IgAN-specific changes.

Reviewer 2 Report

The authors reported the elevation of sCD40L and CD31, but not other cytokines tested, in a relatively large cohort of IgAN patients. In addition, sCD40L and CD31 levels are associated with lower proteinuria. 

This study is mostly exploratory. The findings, although of interest, do not connect to existing mechanisms, such as gd-IgA1 or complements. The connection between sCD40L and CD31 is unknown. Importantly, there are much fewer control cases, and the controls are of different age groups (ADPKD, healthy), and age could also affect cytokine levels. In addition, the elevation of these cytokines is correlated with milder disease, which is counterintuitive, and thus requires more analyses and experiments to consolidate the findings. In the literature, sCD40L level declines as renal function decline. The most likely explanation is, the associations between sCD40L and lower UPCR probably is confounded by the lower level of renal function in patients with high UPCR, thus lower sCD40L. 

Some language issues remain scattered throughout the manuscript. For example, “extremely increased“ is not an appropriate expression. “Multiple regression analysis” should be “multivariable-adjusted regression analysis”. 

I encourage the authors to investigate the CD40 expression on mesangial cells in their biopsy specimens. Currently, the results have not yet provided important insights into the field of IgAN. 

Reviewer 3 Report

Tanaka et al. report that serum sCD40L and Il31 are associated with an early marker of inflammation in IgA nephropathy. This is well-designed clinical study and suggests an important early clinical marker for the management of IgA nephropathy. I have minor comments on this work.

1. The authors use a control group such as ADKPD or healthy control. However, the number is relatively small. Therefore, the authors need to reevaluate the statistical significance.

2. In figure 2E, why did you evaluate relationship between serum IL-31 and sCD40L levels? Because Il-31 and sCD40L are independent factors for early inflammation of IgA nephropathy, what is the clinical meaning of these two serum markers? The authors need to discuss details. 

Round 2

Reviewer 1 Report

1. Please provide evidence that the number of patients in the igA nephropathy and comparator group is statistically adequate.

2. It is an important result to confirm the elevation of two important cytokines, but I wonder if the results of other primary kidney diseases are unique to Ig nephropathy.

3. The evidence for judging it as a prognostic factor for early IgA nephropathy seems to be lacking in the results. Please provide reasons.
